# The Effect of Visual and Interactive Representations on Human Performance and Preference with Scalar Data Fields

Han L. Han*
University of St Andrews

Miguel A. Nacenta†
University of Victoria

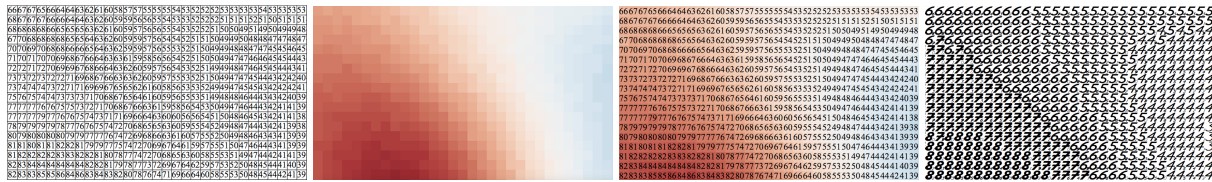

Figure 1: Scalar-field representations in our study: a table of digits (Digits), a red-blue static diverging color scale (Color), a color scale with digits (DigitsColor) and FatFonts. An interactive technique (Tooltip) not shown. The examples are excerpts from larger fields (see supplementary materials for all complete fields).

## ABSTRACT

2D scalar data fields are often represented as heatmaps because color can help viewers perceive structure without having to interpret individual digits. Although heatmaps and color mapping have received much research attention, there are alternative representations that have been generally overlooked and might overcome heatmap problems. For example, color perception is subject to context-based perceptual bias and high error, which can be addressed through representations that use digits to enable more accurate value reading. We designed a series of three experiments that compare five techniques: a regular table of digits (Digits), a state-of-the-art heatmap (Color), a heatmap with an interactive tooltip showing the value under the cursor (Tooltip), a heatmap with the digits overlapped over it (DigitsColor), and FatFonts. Data analysis from the three experiments, which test locating values, finding extrema, and clustering tasks, show that overlapping digits on color (DigitsColor) offers a substantial increase in accuracy (between 10 and 60 percent points of improvement over the plain heatmap (Color), depending on the task) at the cost of extra time when locating extrema or forming clusters, but none when locating values. The interactive tooltip offered a poor speed-accuracy tradeoff, but participants preferred it to the plain heatmap (color) or digits-only (Digits) representations. We conclude that hybrid color-digit representations of scalar data fields could be highly beneficial for uses where spatial resolution and speed are not the main concern.

**Index Terms:** Human-centered computing—Visualization—Visualization techniques—Treemaps; Human-centered computing—Visualization—Visualization design and evaluation methods

## 1 INTRODUCTION

Whenever scalar measurements are collected spatially in two+ dimensions over a defined spatial area, these are commonly stored and represented as scalar data fields (SDF). Scalar data fields are pervasive in scientific and engineering applications ranging from measurements of infrared radiation over a patch of the night sky to the density of probabilities of electrons around an atom. Visualization of SDFs using color-based representations (heat maps) is common, and generally considered to help viewers perceive different aspects of the SDF efficiently. For example, color can help viewers to pre-attentively direct their attention to areas of interest (e.g., areas with high or low values) or grasp the general structure of value distribution. A large corpus of research on color scales has tried to maximize accurate perception of color [42] while sometimes trying to minimize the appearance of artefacts that are due to the representation (e.g., color bands). However, even the best and newest proposed color scales present challenges for the representation of SDFs. For example, viewers can find it difficult to accurately map a perceived color to a specific value of a continuous (or finely discretized) variable, even when there is a good legend. Consider also that the color around a specific value, which itself depends on the surrounding values, can bias the perception of that value (this effect is called simultaneous contrast [31, 47]). Finally, it is difficult to hold arbitrary colors accurately in memory.

In many cases and domains these problems will affect the usefulness of the visualization and determine whether the viewer will require further access to the underlying data. For example, certain scalar values might be semantically important, even if differences with similar values are very small (e.g., energy thresholds of different levels in atomic physics), or if it is a matter of fairness (e.g., finding the precise location of the maximum when there are several competing local maxima).

Available techniques to address the problems of heat maps include a cursor-controlled tooltip that renders the corresponding value's digits (if the media is interactive [30]). Alternatively, if the spatial resolution of the data is not very large or if the spatial subsampling is not very important, the data can be doubly encoded with static digits overlaid on a heat map, or through a property of the digits themselves (i.e., the amount of ink of the digit, in FatFonts [35]). The final alternative is to just go back to the data table to read and compare values represented as digits.

Despite the potential impact of these alternative techniques to display SDFs, there is only sparse research on them (e.g., [30]). Moreover, there is an implicit assumption that color-based representations are better than plain tables with digits; yet, there is no empirical evidence of this, or of the magnitude of the differences.

In this paper we provide new evidence from a controlled lab study that compares four techniques: a popular state-of-the-art color scale, conditional formatting (digits on top of that same scale), an interactive tooltip, and Fatfonts to a baseline table-of-digits representation. The study addresses three common tasks that are relevant for scalar data fields: locating values, finding extrema, and delimiting clusters of values. The study was carried out on a large display. The main

*e-mail: han.han@lri.fr
†e-mail: nacenta@uvic.ca

contribution of the paper is the trade-off emerging from the analysis, which allows designers to make evidence-informed decisions when choosing a suitable visual representation for SDFs. The results show that heatmaps can be dramatically improved in accuracy by adding digits, at the cost of additional time in two of the tasks. The results also refute earlier results that claim superiority of FatFonts for several tasks and measures [30]. Additionally the results confirm the lack of accuracy of tooltips despite the added extra time due to interaction, and quantify the accuracy cost of using color only and the time cost of using digits only.

## 2 RELATED WORK

Our research draws from work in the areas of color scales and mappings, text-based graphical representations and tabular data visualizations.

### 2.1 Color Scales

Researchers have long sought to enhance our understanding of color scales [34, 42]. Among different color scales, Brewer et al. [12] empirically compared diverging, spectral and sequential schemes and found that diverging schemes produce more accurate retrieval than both spectral and sequential schemes and are better for cluster perception than sequential schemes. Moreland's blue-white-red diverging colormap [33] has been used in many visualization applications e.g. [23]. Empirical studies of color maps provide evidence that blue-red continuous diverging colormaps decrease misinterpretations [44] and is the best color choice when utilizing red to highlight the regions of greatest interest [7]. Limitations of color scales have also been identified. First, color and greyscales are limited in the number of levels that viewers can reliably distinguish [47]. Second, for large data sets that require an accurate identification of exact values, color scale visualization can be cumbersome. This can be addressed combining color scales with legends or an interactive tooltip (e.g., as in [1]) [49].

Despite these limitations, color scales are still commonly used for scalar data visualization. We selected for our experiment a best-of-class color scale compatible with the current understanding of color mappings discussed above: a continuous red-blue diverging color scale.

### 2.2 Text-Based Graphical Representation

Text-based graphical representations employ both text and numbers to represent data [10]. For example, Brath et al. [9] extend stem and leaf plots with font attributes. TopeText [50] uses text position to visualize text data for multi-scale spatial aggregates. Other techniques such as tag-clouds [5] and Wordtree [48] encode frequency data with font size and weight. But none of above approaches focus on quantitative scalar fields.

FatFonts [35] double-encode quantitative data through digits, which can be read, and by varying the amount of ink that each digit takes. The amount of ink (or black pixels) of each number is proportional to its numeric value which, due to visual aggregation (similar to stippling), allows the viewer to use the visualization in two different ways: as a global image and as a table of digits. A hybrid technique related to this was presented by Isenberg et al. [25]. Much more common, but also related in approach, is the conditional formatting feature of most modern spreadsheets, which allows to change the color of a cell according to their values.

Very few studies have examined the efficacy of this approach. Chang et al. [14] conducted a controlled study comparing eight encoding techniques to represent edge weights in adjacency matrices. The results show that in an edge-weight comparison task FatFonts are fastest and most accurate, and slowest in a cluster-weight comparison task. Note, however, that adjacency matrices are not exactly like scalar data fields, since proximity of cells does not represent spatial sampling of values as in SDFs. Additionally, the study only tested six levels and did not compare color scales.

Manteau et. al [30] empirically compared FatFonts with other color scale representations for small scalar data fields. They found that FatFonts offer better speed and accuracy for reading and value comparison, and higher accuracy for the extrema finding task. They also found that interactive tooltips were neither fast nor accurate compared with FatFonts. Our work complements this work in several ways. First, their study did not compare other double encoding techniques (e.g., Excel's conditional formatting); Second, they did not include a digits-only baseline condition to measure the effect of color; Third, the color scales that they used were not designed for continuous data and are not currently considered state-of-the-art; and Fourth, they tested at least one task (reading a highlighted value) where the spatial nature of the field is not very important. In our study we share one of the tasks with their design (extrema finding) but add a value location and a cluster delimiting task that offer a better coverage of realistic atomic tasks.

### 2.3 Tabular Data Visualization

In our study we compare heat maps and other representations of SDFs to a static spatial tabular arrangement of digits. Although we did not include sophisticated interactive tabular techniques in our study (partially because they often apply to categorical rather than SDF-like spatial distributions), we acknowledge that there is much work in this area that might be applicable to SDFs in the future, and therefore provide a brief sample of work.

Early research on tabular data visualization focused on interaction techniques. For example, Table Lens [39] uses a fisheye to visualize and explore large tables, where numerical rows and columns can be interactively compressed into visualizations. FOCUS [43] lets users collapse adjacent table cells with the same values. Other recent work has explored the visual encoding of tabular data. For example, BERTIFIER [36] implements eight types of visual encodings: text, grayscale, circle, dual bar chart, bar chart, line, black and white bar chart, and average bar chart. Chang et al. [14] compared eight similar encoding techniques for adjacency matrices.

## 3 EMPIRICAL STUDY

We designed, executed, and analyzed an empirical study to find out evidence-based answers to the following questions:

**Q1** Which SDF representations work best and for which tasks?

**Q2** What is the cost of not using visual variables?

**Q3** What is the cost of not using symbolic representations (digits)?

**Q4** Can interactivity (a tooltip), provide the benefits of symbolic representations without the drawbacks?

To keep the scope manageable, we selected a set of representative techniques and tasks, described by the following sections. Further, we describe the elements common to all sub-experiments.

Although screens with similar pixel counts are available in much smaller sizes, we decided to test different techniques on a large screen because we consider it the state-of-the-art of display for scalar data due to the large physical size. Previous research has shown that physical navigation of a large display has benefits [26, 27]. This has implications for the generalizability of the research that we discuss further in Section 9 (Limitations).

### 3.1 SDF Representations

We consider four SDF representations that we judged representative of current practice and the state-of-the-art, and additional baseline (digits). We refer to these as "techniques" from here on.

---

[1]`https://developers.google.com/chart/interactive/docs/gallery/geochart`

### 3.1.1 Digits

The scalar field is represented only by a grid arrangement of the digits representing the value under their location (see Figure 1, left). Because scalar values are represented exclusively through digits (purely symbolic), it does not afford overview (it is difficult to extract a general distribution of values at a glance, since every data point has to be interpreted individually) and there are no pop-out effects [46].

Despite of the obvious drawbacks, this representation is not uncommon because it requires no translation from raw data other than the appropriate arrangement of its rows and columns. This technique represents the default position of not adding any visual elements to the scalar field.

Tables in our experiments are displayed with two digits per cell and with thin lines separating each cell. Digits that form the same two-digit number are closer together, although the thin lines between cells might also somewhat facilitate grouping.

### 3.1.2 Static Diverging Color Scale (Color)

Adding color variation that depends on the value of the scalar field is likely the most common way of representing scalar fields. The mapping between values and colors is determined by the range of the data and a choice of color scale. The merits of different scales are discussed in Section 2.1 above. Diverging scales have been recently found to be among the best for accuracy [7, 12, 44]. In our study, we used the blue-red diverging color scale implemented in D3[2] [8] (see Figure 1, second from left). D3 uses uniform one-dimensional b-splines to interpolate in RGB color space to convert a discrete color scale into a continuous one[3], which ensures perceptual uniformity of the color. The discrete color schemes being converted are from ColorBrewer[4] [21]. ColorBrewer color scales (including diverging color schemes) have been used in previous studies [7, 12]. This choice also has the advantage that it is not significantly impacted by the most common non-typical vision anomalies.

### 3.1.3 Diverging Color Scale with Tooltip (Tooltip)

When the scalar field is represented through interactive media such as a touch screen, a tooltip could be added to provide viewers with a digital representation for a particular point in the field at their request. In our implementation, the number tooltip appears directly above and left of the tip of the pen. For comparability, the digits have the same size as in the Digits technique.

### 3.1.4 Diverging Color Scale with Digits (DigitsColor)

Instead of showing digits on demand, it is possible to simply overlay digits on top of a field represented using a color scale. This approach makes the image look busier but the digits are still legible, and the color is still visible behind the digits. This is commonly used to represent confusion matrices and is easy to reproduce through the *conditional formatting* feature of spreadsheet software such as Microsoft Excel. Our implementation of this technique simply overlaps the numbers of the Digits technique over the color of the Color technique.

### 3.1.5 Fatfonts

The FatFont technique relies on a manipulation of the digits so that they double-encode the number underneath through the graphical parameter of amount of ink (see description in the Related Work). Like DigitsColor, this approach is a hybrid that uses both symbolic and graphical elements to encode the scalar field's values, with the difference that in Fatfonts the two aspects are integrated into

---

[2] https://d3js.org/
[3] https://github.com/d3/d3-scale-chromatic
[4] http://colorbrewer2.org

Figure 2: Numbers 90, 81, 72, etc. in the study's FatFont variant.

the same visual object (the digit), whereas in DigitsColor the two mappings simply overlap.

We selected a state-of-the-art FatFont variant that is slightly different from the original versions by Nacenta et al. [35]. Instead of putting the second digit (second order of magnitude) inside the first one as in the original versions of FatFonts, in this variant the second digit, which is still $1/10^{th}$ of the area, appears to the right of the first one (see Figure 2). This might make the reading of FatFonts more familiar because the reading of a two-digit number proceeds from left to right (instead of from outside to inside) and is, otherwise, very similar to the original one.

## 3.2 Task Selection

The visualization literature is rich in task taxonomies (e.g., [11, 22, 41]). We selected three tasks that are likely to underline a large number of other more complex tasks; to be relevant for this kind of data and to be informative about the differences between the different technique alternatives described above. The tasks that we selected are also commonly used in evaluation of color scales [7, 12].

### 3.2.1 Locate Value (locate)

A viewer might be interested in locating where a specific scalar value appears. This is important when particular scalar values have specific significance (e.g., critical values or boundaries), or when the existence or absence of particular values is important. The task can be formulated precisely as follows: given a particular value in the scale (the input), locate a position in the 2D plane where the scalar field has that specific value (the output).

This task appears as *searching* in Sedig and Parson's classification [40]. Brehmer and Munzner [11] also identify this as a common visualization task. It also relates to the concept of *reference and characteristics* in Natalia and Gennady Andrienko's book [3]: "Locate is a search task when the identity of search target is known but location unknown."

### 3.2.2 Extrema Finding (extrema)

A viewer might be interested in finding one or two of the limits of the range of values that a scalar field covers (i.e., the extrema). This might be important when maximizing or minimizing outputs or when the dynamic range of a scalar field is valuable information itself. This task only requires as input whether the sought value is the maximum or the minimum.

Finding extrema is one of Amar and Stasko's low-level visualization tasks [2] and was used as a benchmark also by Manteau et al's characterization of small scalar fields [30].

### 3.2.3 Cluster

A viewer might be interested in regions based on the properties of the scalar values, rather than only on specific points. For example, areas of the field where values are above or below a certain threshold. Speed and accuracy in this type of task might be important for applications where identifying regions of interest is relevant; for example, topographic locations below a certain height (which might get flooded), or areas of a scan representing a different kind of tissue.

In this task, participants have to trace a contour that delimits a region of values above or below a certain threshold (the input). This task serves as proxy for the perception of such areas in scalar fields. Several taxonomies include related tasks, such as "cluster" in Amar et al.'s [2] and "identify clusters" in Lee et al. [29].

### 3.3 Participants

We recruited 30 participants from the local university. Five participants were discarded due to problems in the data transfer (2 participants) and due to an error in the counterbalancing (3 participants). The analysis therefore includes 25 participants (9 female, mean age 25). Participants were self-screened for non-typical color vision. Participants were volunteers who did not receive compensation for their participation and provided written consent as specified by the local ethics committee, which approved the study in advance. Responses from a demographic questionnaire indicated that they had a variety of backgrounds (finance, physics, art-history, administrative staff).

### 3.4 Apparatus

All tasks took place in an indoor area with controlled noise and illumination. A vertically mounted Surface Hub 84" display with UHD (3840 x 2160 pixels) resolution was used for input and output. The display is 46.12" x 86.7" (117.15cm x 220.29cm) in external physical size and is mounted on a stand that keeps the lower edge of the display 50 cm from the floor. We discuss the consequences of choosing a large display as apparatus in Section 9 (Limitations).

Participants used the display's pen and touch input (depending on the tasks). The experiment's software is custom-built in node.js for the Google Chrome browser and uses D3.js to display the visualizations. Participants stood in front of the display and were free to move or kneel in its proximity (see Fig. 3).

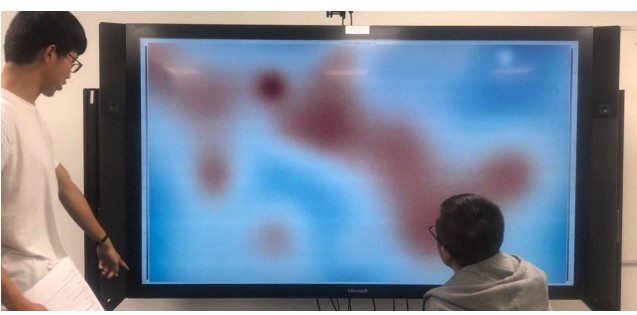

Figure 3: Experimental apparatus

In all experiments, the display showed scalar fields of 190 (horizontal) by 108 (vertical) scalar values. Each data point was represented with a square of 20x20 pixels in size. This guaranteed the legibility of digits while retaining a large enough data field (see further discussion about resolution vs. accuracy in Section 8). We generated synthetic data fields for each experiment (see Section 3.6).

### 3.5 Common Procedure

All participants carried out three sub-experiments, one for each of the tasks selected (locate, extrema, cluster), in the same order. Within each experiment they saw the five techniques always in the same order, assigned to each participant in advance and counterbalanced across all participants using a Latin square. The first time participants encountered a technique (during the locate task), they received a live demonstration of the technique. For each task they received training on how to perform the task. Participants were encouraged to ask questions.

In all experiments there were training trials, which we do not include in the analysis. At the end of each experiment participants completed 7-point Likert scales on their perceived speed and accuracy with the techniques and ranked them in terms of preference. The experiment took approximately two hours to complete; participants took short breaks between tasks and techniques.

In each trial, participants first saw a blank screen with the instructions for the trial (e.g., "Locate target value: 93"), then the field

was loaded, although not yet visible, and participants tapped on a button to start before they saw the visual representation of the SDF. Then they carried out the trial task.[5] Completion time was always measured between the last button tap (when participants first saw the full SDF) and the completion of their answer. Techniques that involve color displayed a large vertical legend along the full left and right sides of the data fields. To avoid noise due to participants forgetting the exact task, the display also showed the instruction at the top of the scalar data field during the trial. If the experimenter noted confusion or an unintended error (e.g., unintended tap on the screen), the experimenter marked the trial as invalid, which automatically added a new identical trial at the end of that trial's block.

### 3.6 Data Fields

We simulated SDFs for the study using MATLAB and R. We sought to simulate fields that: a) are near continuous[6], which is the case for most examples of use that we have encountered (e.g., in Astronomy)[7]; b) span most of the visualization range; c) contain noise; d) are sufficiently complex (i.e., are relatively dense in features) and; e) are comparable to those use in previous studies (e.g., [30]). Fields were created by adding and subtracting 2D Gaussian features from a blank field, shifting and scaling the ranges, and adding white noise at the end. Depending on the experiment, we also selected SDFs that would make the task unambiguous (e.g., fields with a single maximum in the extrema task). The generating and filtering code, as well as the actual data fields used are provided in the supplementary materials.

### 3.7 Data Analysis Approach

We take a Bayesian analysis approach with suitably naive priors for each of the fitted parameters. We ran Markov-Chain Monte Carlo (MCMC) simulations through the JAGS 4.2.0 sampler [37] programmed in the R 3.6.0 programming environment [38] (the scripts are provided in the supplementary materials). The number of simulations was adapted for each test to yield always an Effective Sample Size (ESS) above 10,000. These analyses allow us to calculate approximations of the probability that each comparison between techniques reflects an actual—non-random—difference, based on the data. We do not report omnibus tests, but instead report the calculated posterior probability $p$ that one technique has a higher value than the other, for all pairs. Note that this $p$ is not exactly the same as the more familiar one used in Null-Hypotheses Statistical Testing; for example, when we report a $p > .999$ this means that, based on the data and the model chosen, the likelihood of technique B having a higher average than technique A is almost 100% (i.e., $p > .999$ is a very conclusive result, just as $p < .001$).

We chose statistical models that correspond to the measure and the task. Those that vary are reported separately in each experimental Section. In all experiments, time is logarithmically transformed prior to fitting a Student-t distribution, which provides an alternative to Gaussian models that is robust to outliers [28]. Time averages in seconds report exponentiation (logarithm's inverse) of the log-transformed averages. Our models do not assume equal variances on either participants or techniques. We do not model interaction between the participant and technique factors (not of interest). As recommended in the literature [28] we used Gamma distribution

---

[5]Screenshots are available in the supplementary materials.

[6]Strictly speaking, our data and representations are discretizations of a continuous scale (100 levels), not truly continuous. This level of discretization is well above the discriminability that most color scales have. For practical reasons we refer to the data and scales in this paper as continuous, even though they are really discretized at a reasonably high granularity.

[7]The techniques that we study are applicable to less continuous types of value grids, such as confusion matrices or Perceptual Kernels [16], but in this paper we consider only fields. See Limitations Section

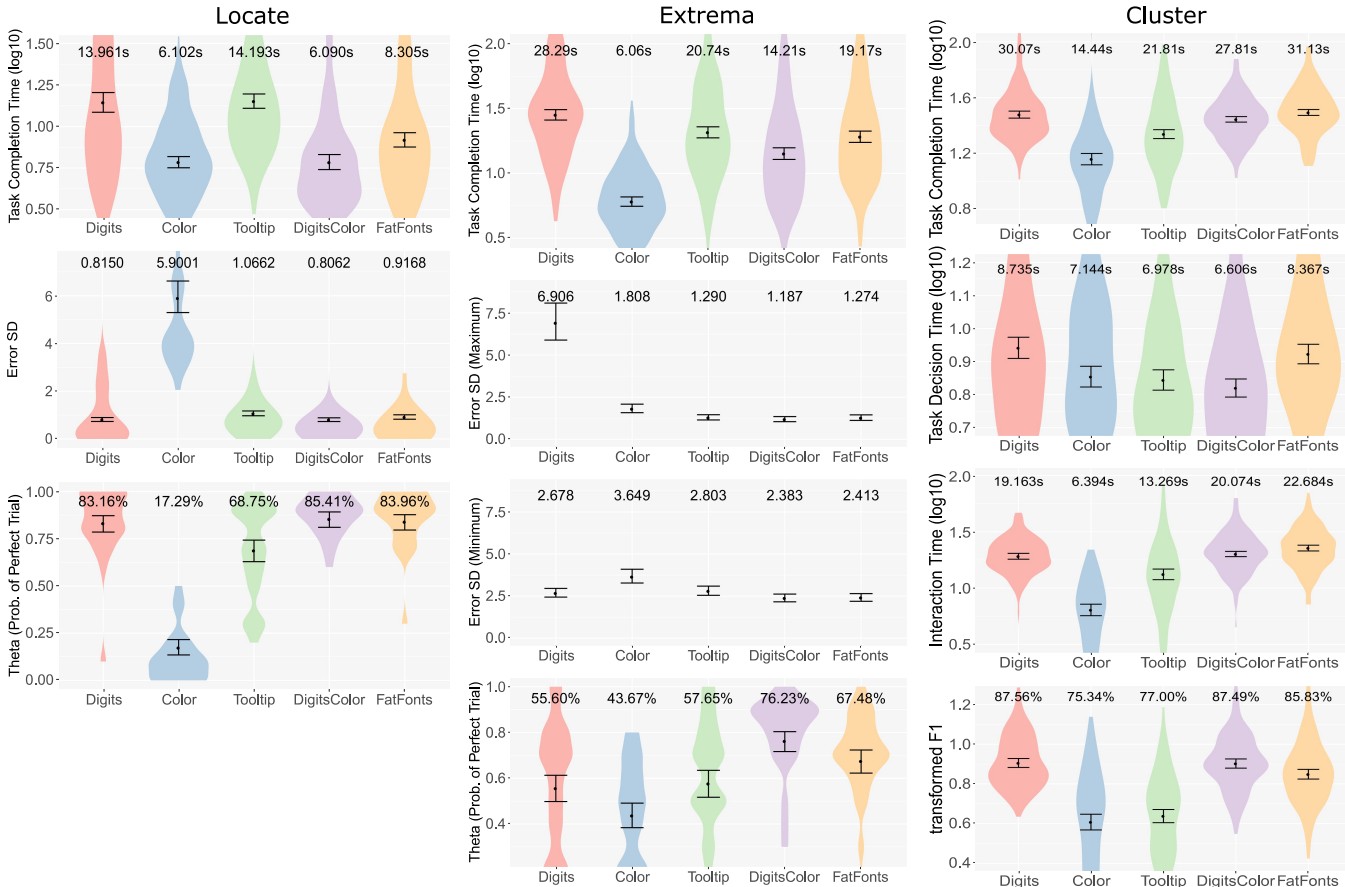

Figure 4: Main measures by techniques for all three experiments: Locate, Extrema, and Cluster (organized by column). Error bars represent 95% Highest Density intervals (HDI) of the estimation of the median, which is displayed also with numeric values at the top of each chart. The colored violins are smoothed representations of the distribution of measured values. Higher values are worse for time measures and error SDs, but best for correctness and F1 scores (the bottom chart in each column). Note that some vertical axes do not start at 0 (for space and clarity).

priors for variance parameters and exponential distributions for the "degrees of freedom" parameter ($\nu$) in the Student-t distribution.

Although this Bayesian approach to analysis is currently more time consuming to design and implement; requires more computing resources to run and is less familiar to readers and other researchers, we chose it over the traditional frequentist approach for multiple reasons: a) it is less prone to some of the serious reliability and interpretation problems that have led to the replicability crisis in Psychology and other areas [15, 17, 24]; b) it allows more adaptable and easier to interpret statistical model fitting than the frequentist alternative (e.g., ANOVA); c) it circumvents the problem of multiple comparisons, and; d) it provides additional information to researchers who want to assess by themselves how the data supports specific claims and use our results in meta-analysis or as priors for their own analysis.

Despite its significant advantages, the output from a Bayesian analysis is, to achieve some of the advantages named above, more verbose. For this reason we ask for additional patience and effort from the reader to read the result table summaries of figures 5 and 6, and the unusually long caption of Figure 5, which guides the reader on how to interpret the results.

For a small but relevant sample of the issues of p-values and frequentist statistics, see Cumming's [15] and Dragicevic's [18] work. Arguments for the use of Bayesian statistics appear in many text books (e.g., [20, 28]).

## 4 EXPERIMENT 1: LOCATE

This experiment measures viewers' ability of finding locations on the screen with a specific value. Participants saw the value (scalar) to search for, tapped on a button, and then provided their answer directly on the screen through finger touch. Participants carried out 15 trials for each technique. Of these, the first 5 were considered training.

### 4.1 Measures, Analysis and Models

For each trial we analyze differences in task completion time, error magnitude (the size of the errors, calculated as the deviation of the answer value from the correct answer value — in output units), and whether the answer was perfectly accurate or not (correctness).

For the statistical analysis of Error, we model the standard deviation of a Student-t distribution as the main parameter of interest, and fit the participant and technique contribution independently. We do not model interaction (see also 3.7). Because some participants made no errors with some techniques, this presents a problem for fitting the models per participant (a Gaussian curve cannot be reliably estimated when its variance is zero). Thus we added a marginal amount of error (1 unit) to every participant's first trial with a technique. This means that our error signal power is very slightly overestimated.

The correctness signal (whether a trial is perfectly correct or not) is modelled as a Bernoulli distribution, where the main parameter $\theta$, fitted per technique and participant, indicates the chance (between 0 and 1) that a given trial is perfectly correct.

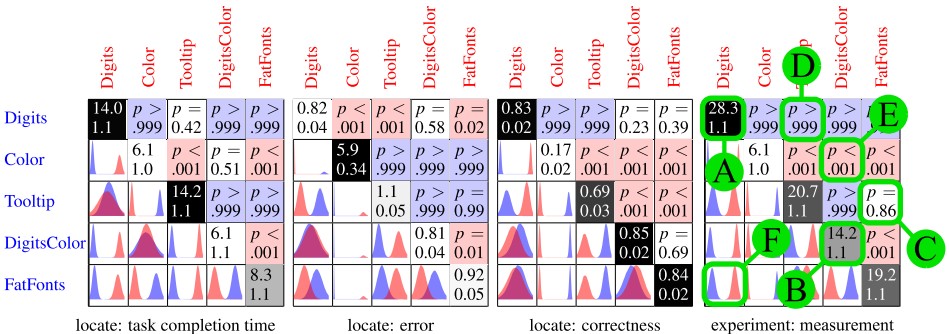

Figure 5: Summary of statistical analyses for Experiment 1: locate. The last figure of the right is a legend to help interpret the conventions of the tables. In each table, which represents a specific measurement within the experiment named at its bottom, each diagonal cell shows the estimated median value of the measurement (top) and the standard deviation of the Bayesian estimation of the measurement that corresponds to that row and column (i.e., a measure of the uncertainty of the estimation). For example, (A) shows that the median for the Digits technique is 28.3, with a standard deviation (SD) of 1.1, all in the same units the measurement (typically seconds for time, proportion for correctness, and output units for error). The background of the diagonal cells is black for the technique with the highest median, white for the lowest and with an interpolated level of gray for the rest (e.g., the values in cell (B)–Tooltip–are a light gray, because the value is somewhere between Digits (A), which is the highest, and Color, in white, which is the lowest). Darker means higher, which is worse for time and error, but better for correctness and accuracy. The cells in the upper-right triangle contain the probability that the row technique (blue) measurement value is larger than the column's (red). Blue shading of these cells indicates that the row technique's value is very likely to be larger than the column technique's value ($p > 0.975$), such as in (D)–Digits >Tooltop, and red shading the opposite ($p(row > col) < 0.025$), such as in (E). This is a proxy for the traditional "p-value" for those unfamiliar with Bayesian statistics, although they are not equivalent. Comparisons not showing strong evidence of the superiority of either technique (i.e., zero is in the 95% high-density interval–HDI–of the difference of plausible values) are left with a white background (e.g., (C)). Cells in the bottom-left triangle (e.g., (F)) show density probability functions of the compared techniques (row technique in blue, column's technique in red). Significant overlap of the curves correspond to probability values in its symmetric cell close to 0.5, i.e., inconclusive comparisons.

## 4.2 Results

Figure 4, left column, displays the main measurements by technique for this experiment. The results of the Bayesian statistical comparisons are summarized in the tables of Figure 5. Digits and Tooltip are similarly slow, requiring more than twice the time as Color and DigitsColor, which are similarly fast (around 6 seconds). FatFonts is slower than the fast techniques using color (about 38% extra time), but closer to them than to the worse performers.

In terms of accuracy, Color's standard deviation is expectedly large, and the rest of the techniques are similarly accurate (although mostly still statistically distinguishable), with levels of noise just approximating one unit. In terms of correctness, participants hit the exact value very rarely with Color, only about 17% of the time on average. Digits, DigitsColor and FatFonts show correctness of between 83% and 85% and indistinguishable from each other; Tooltip lags behind at 69%.

The subjective participant ratings (Figure 8) indicate that participants are able to perceive the objective differences in speed and accuracy, with one exception: Tooltip compares more favorably to Digits, Digits, and FatFonts than would correspond from the objective measurements. Preference rankings show that DigitsColor is most preferred, Digits and Color least and, in the middle, Tooltip is preferable to FatFonts.

## 4.3 Experiment 1 Discussion

For this task, Digits is accurate but slow and Color is fast but inaccurate. DigitsColor is both fast and accurate, and Fatfonts is also accurate, but somewhat slower than DigitsColors. The Tooltip has no redeeming qualities, since it joint slowest and not as correct (by 16% points). The clear winner for this task is therefore DigitsColors.

The results offer several applicable findings. First, the error measure of the Color technique gives us a useful approximation of the expected error magnitude when only color is used to represent a scalar field. Even with the selected state-of-the-art scale, the error is large; our direct measurement indicates that only 18% of the time

the selection is accurate with $\frac{1}{100^{th}}$ unit precision. If we assume a nearly-gaussian model, our fit values indicate that the average error (the absolute difference between the value of a selected pixel and the sought value) will be 4.7 units, close to 5% of the full output space. However, this measurement should be taken only as an initial approximation, since error magnitude is likely to depend on the region of the scale and the distribution of errors might be irregular close to the boundaries, due to floor and ceiling effects.

Second, the interactive Tooltip performs poorly in speed because introducing interactivity increases the expected delays. In our task and setup, Tooltip's performance is equivalent to having no color to guide viewers to the region where the sought value might be (i.e., with Digits only). However, the equivalence in speed between these two techniques is probably just an artifact of our setup, since the additional required time comes from two different sources: in Tooltip from interacting with a region to get the right value, and in Digits from scanning the overall table to find areas where the value might be.

Third, DigitsColor and Color are equivalently fast. This suggests that the overlay of digits does not interfere with the initial search, which is good news for the integrated use of color and digits in visualizations. However, it will be important to validate the result with different fields and screen sizes in future studies, since the sub-tasks performed with both techniques have slight differences (e.g., Color requires looking at a legend).

Fourth, FatFonts shows a slowdown compared to DigitsColor. We have considered several possible non-exclusive explanations for this: a) FatFonts might be unfamiliar and require extra effort to decode compared to the regular numbers in DigitsColor; b) *Amount of ink* might not be as effective at conveying the spatial distribution of the values as color, and; c) The shapes of the digits, which sometimes form visual artefacts, might interfere with the reading or the general search of regions of interest.

Fifth, preference rankings seem to penalize techniques that are clearly worse at either speed or accuracy (Digits and Color, respectively), and select the objectively fast and accurate DigitsColor as

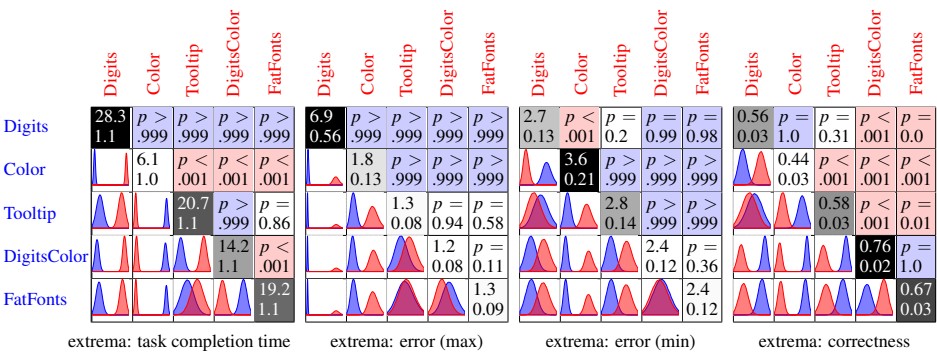

Figure 6: Summary of statistical analyses for Experiment 2 (extrema)

most preferred, as expected. Perhaps surprisingly, there is a slight preference for Tooltip over FatFonts despite the clear time and correctness advantages of the latter.

## 5 EXPERIMENT 2: FIND EXTREMA

This experiment is designed to measure viewers' ability to find the global maximum or minimum within a SDF. Participants saw an indication of whether they had to search for the maximum or the minimum, tapped on a button, and then provided their answer through finger touch. Participants carried out 14 trials for each technique (half maxima and minima). Of these, the first 4 were considered training. SDFs used in this task only had one correct answer (a global maximum or minimum) but multiple local extrema.

### 5.1 Measures, Analysis and Models

We followed the same analysis as in Experiment 1, except that minima and maxima errors were analyzed separately due to their asymmetry. Since maxima and minima errors can only go in one direction, we fit a half Student-t distribution (truncated at the mean) instead of the regular two-tailed one. This is also the reason why Figure 4, middle column charts do not show violin plots for the Error measurements.

### 5.2 Results

Figure 4, middle column, shows the aggregated values of the measurements for the extrema experiment. Figure 6, bottom row, summarizes the statistical comparisons. In terms of speed, the Color technique is fastest (6 seconds), followed by DigitsColor (more than twice as slow), with Tooltip and FatFonts next (at the 20 seconds mark) and with Digits last (28 seconds).

Accuracy measurements follow almost the same pattern, except that Digits has a severely inflated error for maxima that is not so for minima. DigitsColor, FatFonts and Tooltip are most accurate for maxima, with Color somewhat behind, and then Digits, with accuracy many times worse. For minima, DigitsColor and FatFatfonts are most accurate (although statistically indistinguishable from each other), followed by Digits and Tooltip, with Color being the noisiest.

A clearer picture emerges with the correctness measure, where DigitsColor has the greatest correctness (76%), followed by Fatfonts almost 10 percent points behind, with Tooltip and Digits around 55% (and indistinguishable from each other), and Color as worst, with 44% correctness.

The subjective measurements in Figure 8 indicate that, consistent with the objective measurements, participants identified Color and DigitsColor as fastest and DigitsColor as most accurate. Preference rankings follow logically from there, with DigitsColor as top, FatFonts as second, Digits and Color as worst, and Tooltip squarely in the middle.

### 5.3 Experiment 2 Discussion

The extrema task shows a similar pattern as in the locate experiment but with several important differences. The best technique is again DigitsColor, which is substantially slower than Color only, but is correct almost twice as often. Color might be acceptable if correctness and error are not an issue and speed is paramount. FatFonts is reasonably close in accuracy and speed to DigitsColor, but worse in both. Digits and Tooltip are both at the bottom of the list, since they are neither fast nor accurate compared to the best.

Unlike in the locate experiment, the accuracy of DigitsColor comes at a cost in time, as evidenced by the substantial difference in completion time compared with Color. The extrema task takes about the same time as the locate task when only color is available, but much longer when digits are available. This is probably because finding extrema with numbers requires keeping a temporary highest/lowest number in memory, and comparing each number to decide whether to update it. In contrast, the locate task only needs to decide whether each scanned value matches the sought value or not, until a match is found. Our data provides a first approximation of this cost: twice the time.

The results also show that digits and color (or ink) are complementary. The Digits technique is less accurate because participants are likely to be stuck in a local extremum; without the ability to see the overall image, jumping between areas of interest is very costly. The Color technique is less accurate for two reasons. Although the overall areas of extremum candidates are easy to access and jump between, the colors in distant areas are difficult to compare because it is hard to keep an accurate color in memory and the surrounding colors are likely to distort perception through the simultaneous contrast effect [31, 47] and possibly other top-down perceptual effects [1]. Even within the same region, it is difficult to ascertain small differences in color between contiguous locations (e.g., 1 or 2 units, which correspond to 1 and 2% of the perceptual space.)

We also see the disadvantage for Fatfonts compared to DigitsColor in speed, but here also in accuracy/correctness. The source of this differences might be the same as speculated in the dicussion of Experiment 1. It is also possible that the smaller second digits in the FatFont variant that we used are harder to read and therefore cause some additional errors.

Except for the Digits high error for maxima, we observe that error standard deviations are higher (double) for finding minima than maxima. This asymmetry replicates results from a previous study [30] and might be caused by non-linearity in perception (e.g., Weber's Law [19] or Steven's law [4]). This issue might be of relevance for future study.

A close look at the maxima trials indicates that sometimes errors were extremely large in the task of finding the maximum, with discrepancies of over 60 units. Since this anomaly only appeared

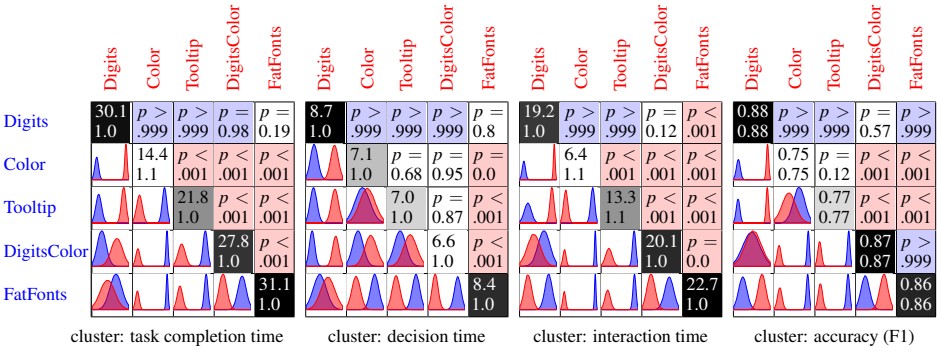

Figure 7: Summary of statistical analyses for Experiment 3 (cluster)

for the Digits technique and technique order was randomized, it is unlikely due to participant confusion of the minimum with the maximum subtask. Another possiblity is that participants, searching for high numbers, confused the first digit 9 in the number with number 0, which are morphologically similar. We suspect that the anomaly does not correspond to a real-world effect, but future study should rule this out. Finally, the subjective preference measurements do not offer any surprise for this task.

## 6 EXPERIMENT 3 CLUSTER

This experiment represents a task where a group of contiguous data points need to be separated from the rest based on their values. Participants were shown instructions of what values form a cluster (e.g., "Inside values above or equal to 73"), then tapped on a button to start the task, and then used the pen to draw a contour consistent with the instruction.

### 6.1 Measures, Analysis and Models

In this experiment we measured the overall completion time of each trial, but also split the trial time between *decision time* (time between start of trial and first contact of the pen) and *interaction time* (time spent drawing the contour). We measured error using the F1-score according to the following formula:

$$F1_{score} = \frac{precision * recall}{(precision + recall) \times 2}, \qquad (1)$$

where precision is the proportion of the pixels in a participant's answer that are part of the correct answer, and recall is the proportion of pixels of the correct solution that are part of the participant's answer. The F1 score provides an appropriate balance between precision and recall. Due to an error in the pixel-counting algorithm we introduced a small error of up to 7 pixels (which were counted twice); due to this error the results are slightly more conservative than they would have been. However, since the error applies to all techniques and is an upper bound, the comparisons made with this data are still valid.

Times were modelled as in previous experiments. To be able to fit the F1 scores (which have a ceiling at 1.0) with Student-t distributions, we transformed the F1 scores using the following formula:

$$F1_{transformed} = -\log_{10}(-F1 + 1). \qquad (2)$$

### 6.2 Results

The three time measurements and the accuracy of this task are displayed in Figure 4, right column, and the statistical tests appear in Figure 7, bottom row. For overall speed, Color is again fastest (14 seconds on average), followed by Tooltip (22 s), DigitsColor (28

s) and with Digits and FatFonts last, but very close by (and statistically indistinguishable from each other). The decomposition of task time shows that decision times are fairly similar across techniques although there are two statistically defined groups: DigitsColor, Tooltip and Color are fastest (between 6.6 and 7.1 s), and FatFonts and Digits form the slower group (above 8.3 seconds). Interaction time tracks fairly closely the overall completion time, of which it is the dominant component.

The F1 accuracy measures shows Digits and DigitsColor as most accurate (F1 scores of 0.87), with Fatfonts closely following (0.85), and Tooltip and Color about 10% points lower. The subjective ratings of speed are surprisingly close to each other, with Color and DigitsColor rated very close to each other, whereas the accuracy ratings correspond to the objective measurements. Subjective preference penalizes Color and Tooltip and highlights DigitsColor as best.

### 6.3 Experiment 3 Discussion

As in Experiment 2, techniques that took longer to complete were also more accurate, and vice versa. Therefore there is a clear trade-off: If speed is the most important criterion, then Color by itself is best; If accuracy is important, then one can get an improvement of 10 percent points in exchange for doubling the time with the DigitsColor technique.

The Digits and Fatfonts techniques follow DigitsColor fairly closely in accuracy and speed, but do not beat DigitsColor in either. The reason for the small advantage in time can be explained by the overall shorter decision time of DigitsColor. This time reflects how long it took participants to be confident enough to start drawing the contour, and is shorter for the three techniques that use color. This indicates that FatFonts, despite using a visual variable to form an image, does not seem to help much with the visual overview, since it showed decision times close to those of the Digits technique (between 1 and 2 seconds slower than the color-based techniques). The exact reason for this difference is difficult to guess from our measurements only. We can speculate that amount of ink might simply be harder to process by the perceptual-cognitive system, or that the perceptual and symbolic aspects of the FatFonts digits might interfere with each other. However, further research is needed to elucidate this, and the size of the SDF might also play a role here.

The Tooltip is, again, not a justifiable compromise. It is a few seconds faster than DigitsColor, Digits and Fatfonts, but slower than Color by about 7 seconds on average and with an equivalently low performance, about 10% lower than the others. This is partially explainable by the somewhat awkward requirement to trace a line while a number appears next to the tip of the pen. We did not implement pen-and-touch input for this because most displays do not allow this double interaction style. Furthermore, having to control two inputs simultaneously adds additional difficulty.

**Subjective Measurements**

| | Locate | | | | | | Extrema | | | | | | Cluster | | | | | | Overall | |
|---|---|---|---|---|---|---|---|---|---|---|---|---|---|---|---|---|---|---|---|---|
| | Speed | | Accur. | | Pref. | | Speed | | Accur. | | Pref. | | Speed | | Accur. | | Pref. | | Pref. | |
| | M | μ | M | μ | Rk | μ | M | μ | M | μ | Rk | μ | M | μ | M | μ | Rk | μ | Rk | μ |
| Digits | 4 | 4.5 | 2 | 2.5 | 4 | 4 | 5 | 5 | 3 | 3.4 | 4 | 4.2 | 3 | 3.2 | 2 | 2.3 | 3 | 3.1 | 4 | 3.8 |
| Color | 2 | 2.2 | 6 | 5.6 | 4 | 4.2 | 2 | 2.1 | 4 | 4.8 | 4 | 3.8 | 1 | 2.2 | 6 | 5.3 | 4 | 4 | 4 | 4 |
| Tooltip | 4 | 3.7 | 2 | 2.2 | 2 | 2.6 | 3 | 3.7 | 3 | 3.4 | 3 | 3.1 | 3 | 3.2 | 4 | 4.2 | 4 | 3.9 | 3 | 3.1 |
| DigitsColor | 2 | 2.2 | 2 | 1.8 | 1 | 1.3 | 1 | 1.9 | 2 | 1.9 | 1 | 1.2 | 2 | 2.2 | 2 | 2.4 | 1 | 1.3 | 1 | 1.1 |
| FatFonts | 3 | 3.2 | 2 | 2.3 | 3 | 2.9 | 3 | 3.5 | 3 | 3 | 2 | 1.8 | 3 | 2.8 | 2 | 2.4 | 3 | 2.8 | 3 | 3 |

| | Loc: Speed | Loc: Accur. | Ext: Speed | Ext: Accur. | Clus: Speed | Clus: Accur. | Resolution | Precision | Req. Interact. | Req. Color |
|---|---|---|---|---|---|---|---|---|---|---|
| Digits | 5 | 1 | 5 | 3 | 5 | 1 | 5 | 3 | 1 | 1 |
| Color | 1 | 5 | 1 | 5 | 1 | 5 | 1 | 5 | 1 | 5 |
| Tooltip | 5 | 4 | 3 | 3 | 2 | 4 | 1 | 1 | 5 | 5 |
| DigitsColor | 1 | 1 | 2 | 3 | 1 | 3 | 3 | 3 | 1 | 5 |
| FatFonts | 3 | 1 | 3 | 2 | 5 | 3 | 5 | 2 | 1 | 1 |

Figure 8: Subjective results and preference rankings for all experiments and global preference (left). Summary of quantitative and qualitative (grey background) results (right). Readings are transformed so that 1 (in saturated blue) means best (first in rankings, most favourable Likert ranking for Speed and Accuracy), and 5 means worse (in saturated red).

Participants preferred techniques that they saw as accurate for this task, leaving Color and Tooltip at the bottom, perhaps also affected by the problems of interactive control discussed above.

## 7 SUBJECTIVE RESULTS

The global subjective preference rankings at the end of the experiment (Figure 8, left table, rightmost column) place the techniques in three clear groups. DigitsColor is most preferred, followed by Tooltip and Fatfonts, and with Digits and Color last.

## 8 GLOBAL DISCUSSION

When considering the three experiments together, the emerging theme is of a clear trade-off. The first six columns of the right table in Figure 8 summarize the quantitative results and the left table summarizes the subjective ratings and ranks. The summary shows that in all tasks DigitsColor was best, or equivalent to best in accuracy and correctness, and shortest, or close to shortest in completion time if we discount the fastest technique (Color). If accuracy is important for the specific application, the DigitsColor representation, which combines static digits and color, is both objectively superior to the rest and preferred by our participants (Q1). However, accuracy comes at a cost in the extrema finding and clustering tasks: about 8 and 13 extra seconds respectively compared to Color, which is always the fastest or equivalent to the fastest technique. This amounts to roughly doubling the task completion time.

Double-encoding the values in the amount of ink of a digit (FatFonts) offered advantages in accuracy over using either color or digits only for the first two tasks, but is not better than using digits and color in combination. This supersedes the results from Manteau et al. [30], which found FatFonts to be the best representation; we now know that this is because they did not compare FatFonts to a hybrid digit+color approach. Overall the FatFonts approach is second to DigitsColor (also known as conditional formatting with a color scale in spreadsheet jargon).

Manteau et al. [30], as all other previous literature that we are aware of, have also hitherto neglected to compare SDF representations to a plain table (Digits). This comparison is useful since it quantifies the time benefit of supporting overview by encoding values through visual variables in data fields. We found that the digits only representation is always the slowest (or equivalent to slowest), and that the differences in times are substantial (5 times slower than Color for extrema finding, and almost double for the other two tasks) (Q2).

Additionally, we provide first estimations of the high perceptual noise of using color alone. Compared with the best digit-based technique, it multiplies by seven the magnitude of the error in the Locate task; increases it by about 50% when finding extrema; results in F1 scores of about 12 fewer percentual points for clustering; and much less likely perfect answers for both the Locate (50 percentual points lower) and Extrema tasks (33 percentual points lower). This answers Q3 and supports long-held beliefs about the interference of

simultaneous contrast effects [47] and the difficulty of using legends for detailed reading of univariate scalar fields. Although these are valuable first steps, the hybrid serial and parallel nature of the tested tasks necessarily imply that varying SDF size and, perhaps, spatial frequency of the scalar data, will also influence completion times and accuracy of the techniques differently. Future experiments will model these dependencies to support better choice representation for SDF data depending on the specific situation.

Our results do coincide with those of Manteau et al. in how poorly the interactive tooltip performed. As they also discuss, one might think that showing digits on demand is, a priori, an efficient way to add precision and protect from the issues of using color only. However, our results show that the interactive tooltip is typically much slower than color, and less accurate and correct than other digit-based representations (Q4). Despite this, participants preferred Tooltip above Digits and Color, and on par with FatFonts, only after DigitsColor in their overall ranks. We speculate that Tooltip's interactivity might allow viewers to engage more directly with the data, in a more "constructive" fashion, which has been shown to increase sense of control and authorship in visualization creation [32].

Despite the clear results for our experiments, we should highlight that there are still situations in which the best performing techniques might not be desirable. Some of these are summarized in the rightmost four columns of the right table in Figure 8. For example, techniques using color might be more affected by variable color perception in the population, and techniques with static digits have limits in the spatial resolution of the underlying SDF that they can enrich[8].

## 9 LIMITATIONS AND FUTURE WORK

Here we make the reader aware of limitations of our results concerning their applicability and generalizability. First, as we detailed in the Empirical Study and Apparatus sections, we decided to run this study on a large display. Our main motivation was to give all techniques the best chance; there is a growing body of evidence that large displays are beneficial in terms of physical navigation and memory (e.g., [6, 26, 27]). Additionally, a larger display — which typically has larger resolution — enabled us to display larger scalar fields. This covers a wider number of use cases for the digits-based techniques because they depend on displays with high pixel counts.

In exchange, our experiment became less representative of the more common analysis situation where analysts sit in front of their laptop or monitor to look at scalar field representations. Nevertheless, we found no obvious reasons to believe that the results would vary much across different display sizes of similar retinal resolutions.

A second related, but distinct, issue is the maximum size/resolution of SDF representations. Digit-based representations require multiple pixels to represent a single scalar value, whereas a color scale can represent a scalar value with a single one. This

---

[8]Nacenta et al. [35] and Manteau et al. [30] provide extended discussions of issues with spatial resolution.

is clearly a limitation for digit-based techniques (also extensively discussed in [30, 35]). However, maximizing spatial resolution might not be the only priority, and the perception of color through the small area of single pixel, especially in a modern high-resolution display, is likely to result in further perceptual issues (e.g., [13]). We suspect that in many SDF use cases spatial resolution is not the dominant factor in their usefulness; yet, this question can only be answered through a systematic investigation of the purpose and usage of SDFs in real scenarios, of which existence we are not aware and which falls outside the scope of this work.

Third, our experiment only tested SDFs which, although very common, exclude similar data types such as confusion and adjacency matrices [14, 45], perceptual kernels [16] and other similar data arrangements. In those, the categorical nature of the data means that values in spatially close cells can vary much more abruptly, resulting in higher spatial frequencies overall that are different from the typically softer variations from spatial data. Although there is already some evidence that digit-based representations are useful for this kind of data as well (e.g., [14]), Future study should verify that the differences that we observed also apply to these other types of data.

Fourth, although we chose techniques that we believe are the best representatives available, the design space of techniques is vast. For example, there might be color scales that can be optimized for some of the tasks that we tested, or alternative tooltip designs that offer a better overview-detail balance. The design space of FatFonts is also large, and we chose a variant different from that in Manteau et al.'s study [30]. Although this could potentially introduce a confound and reduce the comparability of both studies, we have no strong reason to suspect such problem. We actually think that the version that we used is more familiar and likely to have an advantage (perhaps only marginal) over the one used by Manteau et al [30].

## 10 CONCLUSION

Two-dimensional data fields are common and are often represented through heatmaps. Although there exists a large body of research on color mappings and how they support different tasks when representing univariate scalar data fields, these comparisons usually ignore the possibility of using digits to represent the values. In this paper we describe a series of three studies that tested five different representations of scalar data fields in three corresponding tasks: locate values, finding extrema, and clustering regions. We found that using a state-of-the-art color mapping without digits resulted in the fastest performance, but also the largest errors. Overlapping digits (the DigitsColor technique) offers more precise results, but increased completion time in two of the three tasks that we tested. Other representations with digits, such as FatFonts, or adding an interactive tooltip that shows the digits "on demand" were either slower or less accurate than the DigitsColor technique. Comparisons with the color-only and the digits-only representations provide a first quantification of the contribution of color and digits to performance when reading scalar data fields. Overall, the results indicate that digit-based representations combined with color can substantially increase accuracy at a relatively small cost in time, and should therefore be seriously considered by designers of visualizations of SDF data where spatial resolution is not the dominant concern.

### ACKNOWLEDGMENTS

The authors wish to thank Ranchao Ou for his help on the initial data analysis, Kevin Pirazzi Maffiola for the insightful discussion. This work is the Master Thesis of the first author and was funded by his parents. The first author is now a Ph.D student in ex)situ team Paris, France.

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
