# OpenReview forum: "The Effect of Visual and Interactive Representations on Human Performance and Preference with Scalar Data Fields"
_graphicsinterface.org/Graphics_Interface/2020/Conference — GI 2020_

### Official Review · AnonReviewer2 · 2020-04-20

**Rating:** 8
**Confidence:** 4

**Review:**

In this submission, the authors present their results of a comparative study on scalar-data fields between 5 different visual representations. The paper is well written (except a few typos mentioned at the end of this review) and easy to follow. I am not an expert in the topic but I would argue that enough related work was covered. The results are interesting and relevant to the community and I would overall argue for accepting the submission.

I nonetheless have a couple of comments and issues with the submission that I detail below.

I particularly appreciate that the authors made their stimulus and data analysis available. I would recommend however to put all files in a safe online repository (such as osf.io) --- instead of relying on supplementary materials ---  and then link to the online repository.

I also enjoyed the bayesian analysis of the results that was quite easy to understand (I would nonetheless remove the 2nd paragraph of page 6, which seems a bit odd in a research paper). I would also argue that the caption of figure 5 should mostly be in the text and not used as a caption. The caption should mostly describe the figure.

It was somewhat annoying that the results for experiment 2 and 3 were not close to the text. Perhaps the authors could envision splitting their figures differently to make it easier to read the figures and the text together.

The authors mention in the limitation “Our main motivation was to give all techniques the best chance; there is a growing body of evidence that large displays are beneficial (e.g.,[23, 22])” but I was expecting there to get a quick summary of what they mean by this.

Minor:
- Make sure all references are in order (e.g., in the introduction, “[32,7]” or “[44,27]”).
- There is a comment from one author left in the manuscript I believe "CUT:that we judged” (page 3, bottom left)
-”preavious” → “previous” (page 3, right)
-”underlie” → “underline” (page 4, left)

---

### Official Review · AnonReviewer1 · 2020-04-21
**Review of The Effect of Visual and Interactive Representations on Human Performance and Preference with Scalar Data**

**Rating:** 6
**Confidence:** 4

**Review:**

This paper reports on a study on scalar data fields (SDFs) comparing 5 techniques (Digits, Color, ToolTip, Digits+Color, FatFonts variant juxtaposed not embedded) through 3 tasks (Locate Value, Find Extrema, Cluster) on a large display in a controlled lab experiment with 25 participants recruited from their university. With various reported metrics (time, errors), the main outcome is that Digits+Color (a table of digits overlaid with a heatmap) is recommended for increased accuracy with a small trade-off in time when spatial resolution is not a constraint.

I would recommend for acceptance of this paper.

Evaluation based on criteria suggested for reviews

Quality

Some claims in the introduction could benefit from references. See details thereafter. The technique named FatFont in the paper is not identical to previous related works: digits are embedded in [26,31], juxtaposed in this submission; so comparing results requires exercising caution.

Clarity

Figure 1 would have gained to clearly show the 5 techniques as named thereafter in the study. Figures 4,5,6 demand some effort to be interpreted, particularly since color mapping is not consistent across Figures: technique (4), column/row (5), value (6). I am not convinced by the need of a statistical method (Bayesian vs Frequentist) that is "less familiar to readers and other researchers" that diverts attention and requires lengthy explanations (Figure 5 caption and disclaimer on page 6).

Originality

This work builds upon previous studies on FatFonts presented at GI'17 [26] by introducing a straightforward baseline in the comparison: Digits (table).

Signifiance

While this is not clearly addressed in this submission, I believe that overlaying heatmaps over tables, the recommended technique overall, also directly brings benefits to presentation in scientific papers.

Comments organized by appearance over the paper

Abstract

I would suggest to align presentation orders and descriptions of techniques in Abstract vs Figure 1 for faster understandability:
"
1) a state-of-the-art heatmap:
2) regular tables of digits,
3) an interactive tooltip showing the value under the cursor,
4) a heatmap with the digits overlapped over it,
5) and FatFonts.
"
vs Figure 1
"
a) digits (table),
b) red-blue static diverging color scale,
c) color scale with digits (conditional formatting)
d) and FatFonts
"
Is the following mapping correct?
1=b?
2=a
3=?
4=c
5=d
Why not use the names of techniques as in the study (Digits, Color, ToolTip, Digits+Color, FatFonts)?

INTRODUCTION

"A large corpus of research"
Could references be cited to support this claim?

"appearance of artefacts that are due to the representation "
Are artefacts also present for techniques using text-based representations?
Marcos Serrano, Anne Roudaut, and Pourang Irani
Investigating Text Legibility on Non-Rectangular Displays.
CHI ’16
DOI:https://doi.org/10.1145/2858036.2858057

"value of a continuous (or almost continuous) variable"
What is an almost continuous variable?

"Available techniques to address the problems of heat mapsinclude a cursor-controlled tooltip that renders the correspond-ing value’s digits (if the media is interactive)."
Any reference? Is it the same tooltip used in [26]?

" The results also refute earlier results about FatFonts being the best representation [26]"
The last sentence of the abstract of [26] reads "The FatFonts technique showed better speed and accuracy for reading and value comparison, and high accuracy for the extrema finding task at the cost of being the slowest for this task." [26] does not claim that FatFonts are the best representation.

RELATED WORK

Color Scales

How does color blindness affect the choice of color scales?

Text-Based Graphical Representation

"Third,the color scales that they used are not currently considered state-of-the-art, or the best current ones for continuous data"
Why? Would you have a proof to support your claim?

EMPIRICAL STUDY

"In our study, we used the blue-red diverg-ing color scale implemented in D3 2"
In addition to footnote 2, why not cite at least one article by the authors of d3?
M. Bostock, V. Ogievetsky and J. Heer, "D³ Data-Driven Documents," in IEEE Transactions on Visualization and Computer Graphics, vol. 17, no. 12, pp. 2301-2309, Dec. 2011.

"ColorBrewer color scales (including divergingcolor schemes) have been used in preavious studies [6, 11].This choice also has the advantage that it is not significantlyimpacted by the most common non-typical vision anomalies"
Not all color scales proposed by colorbrewer.org are colorblind safe. Try to tick/untick the related checkbox in the website.
In addition to footnote 4, why not cite the article by the authors of Colorbrewer referenced in the information popup on Number of data classes in their website?
ColorBrewer: An online tool for selecting color schemes for maps. The Cartographic Journal 40(1): 27-37. 2003
http://doi.org/10.1179/000870403235002042

Please typeset URLs in footnotes 1,2,3,4 correctly with \url{} or \href{}{} as the document already uses LaTeX hyperref according to document properties.

"We selected a state-of-the-art FatFont variant that is slightlydifferent from the original versions by Nacenta et al. [31].Instead of putting the second digit (second order of magnitude)inside the first one as in the original versions of FatFonts,in this variant the second digit, which is still1~10thof thearea, appears to the right of the first one (see Figure 2)."
The FatFont variant is interesting, but in that case comparison with previous work does not apply.

" such as "cluster" in Amar and Stacko’s [2] and "identify clusters"in Lee’s [25]."
[2] has 3 authors (not 2) and [25] has more than 1.


"We recruited 30 participants with a variety of backgrounds(finance, physics, art-history, administrative staff) from thelocal university. "
Is the variety of backgrounds desired or emergent from recruitment?

"If the experimenter notedconfusion or an unintended error (e.g., unintended tap on thescreen), he marked the trial as invalid, which automaticallyadded a new identical trial at the end of that trial’s block."
Plural "they" vs "he" would be more inclusive for all "25 participants (9 female)".

"We simulated SDFs for the study using MATLAB and R."
Why both environments and not just one?

"6 Strictly speaking, our data and representations are discretizationsof a continuous scale (101 levels)"
Why 101?

"We ran MCMC simulations through JAGS 4.2.0 [34]"
It would be great to explain both acronyms at their first occurrence in the document here.

"(variance cannot be reliably estimated when it is zero)"
In that case variance is then reliably equal to zero? I guess that the formulation of this sentence needs editing.

"a) it is less prone to some ofthe serious reliability and interpretation problems that haveled to the replicability crisis in Psychology and other areas"
Any reference to support this claim?

"For this reason we ask for additional patienceand effort from the reader to read the result table summaries offigures 5, 7, and the unusually long caption of Figure 5, whichguides the reader on how to interpret the results."
Figure 5 does not seem to be cited elsewhere in the paper other than here in this disclaimer. When should readers check Figure 5?

EXPERIMENT 1: LOCATE

"Tooltip is preferable to FatFonts"
To avoid overgeneralization I would suggest rephrasing into: "Participants preferred Tooltip over FatFonts".

Figure 5: "Darker means higher, which is worse for time and error, but better for correctness and accuracy."
So "lighter is better" for subfigures 1 and 2, but "darker is better" for subfigure 3. This seems prone to confusion.

EXPERIMENT 2: FIND EXTREMA

"For minima,DigitsColor and FatFatfonts are most accurate (although statisically indistinguishable from each other)"
How are both statistically indistinguishable?
Error SD (Minimum) values are different between both in Figure 4, row 3 col 3.

I have an issue with the lack of consistency in assignment of colors across Figures.
- Figure 4: colors (red, blue, green, violet, yellow/orange) are categorically mapped to techniques
- Figure 5: colors are mapped to rows (blue) and columns (red) for pairwise comparison
- Figure 6: colors are mapped quantitatively to values between 1 (blue) and 5 (red)

Meta question: how do the results of the study described in the paper inform how Figures in the paper could be optimized so that readers can easily locate values and extrema across metrics, and cluster techniques to better understand results of the study?

EXPERIMENT 3 CLUSTER

"We did not implement pen-and-touchinput for this because most displays do not allow this doubleinteraction style."
Large displays? Because many displays, like Apple iPad + Pencil, the Microsoft Surface series and Wacom Cintiq series support pen and touch.

LIMITATIONS AND FUTURE WORK

"there is a growingbody of evidence that large displays are beneficial (e.g.,[23,22])."
I have a few more reference to suggest:
- Xiaojun Bi and Ravin Balakrishnan.
Comparing usage of a large high-resolution display to single or dual desktop displays for daily work.
CHI ’09
DOI:https://doi.org/10.1145/1518701.1518855
- Fateme Rajabiyazdi, Jagoda Walny, Carrie Mah, John Brosz, and Sheelagh Carpendale.
Understanding Researchers’ Use of a Large, High-Resolution Display Across Disciplines.
ITS ’15
DOI:https://doi.org/10.1145/2817721.2817735

---

### Official Review · AnonReviewer3 · 2020-04-22
**Review paper 70**

**Rating:** 8
**Confidence:** 4

**Review:**

This article compares five representations of scalar fields: number table, fatfonts, color map, color map+tooltips, color map+numbers. The experiment consisted of three tasks: locate a value, find an extrema, and identify a cluster, on synthetic data. It involved 25 participants conducting the tasks on a large display (84"). Based on a bayesian analysis of the data, the authors discuss the trade-offs between accuracy and speed involved with the techniques.

The paper is clearly written and nice discusses the related work (albeight too cursorily when it comes to data tables). The analysis method is interesting and appear to be solid. Like in many experimental studies of the sort, I was left wondering to which extent some simple design improvements on the techniques would not have changed the outcome of the experiment. And the techniqu studied are not particularly original.
This does not dimish the results, but rather I would encourage the author to further frame their experimental questions in real-life visualization work, and discuss them in term of design choices. Alternatively, broad, generic questions such as  symbolic vs. visualization can be asked, but in such a case, a bit more modeling or theorization would be expected, in order to draw some lessons from the experiment. We are left somewhat in between precision generality and precision. On one hand, the paper provides some useful insights on the pros and cons of the techniques for differents tasks, but the techniques are somewhat rough (the tooltip technique is not very subtle and much richer forms of interaction could be imagined, color inversion of text in the color+digits could help readability, the number table layout could be improved for better legibility...) On the other hand, we are left with the experimental results without a model of speed/accuracy tradeoffs that could help pick the best representation, or some explanations on why some techniques perform better in one task.

I commend the authors for their analysis. I am not expert enough to assess the quality of the bayesian analysis, but the presentation is clear, and I could follow from beginning to end. The supplementary material is detailed and useful to reproduce the analysis (I haven't tried to run the notebooks though).

Regarding the presentation, I would suggest to split the figures so that the figures associated to each task appear in the relevant section. On figure 4, it feels like starting all y axes at 0 would give a better sense of the distribution of the results. Regarding fig.5 the caption is not legible, as VIS/HCI researchers I hope we can do better than warn the reader that it's going to be challenging.
-> why not annotate the figure directly, rather than adding an indirection (A, B, C...) ?
-> why not give a bit more salience to the interesting results and "fade" the other ones ?

---

### Meta-Review · Area_Chair1 · 2020-04-23

**Recommendation:** Accept
**Confidence:** 4

**Metareview:**

This submission has received the following scores:
- Average Rating: 7.33 (Min: 6, Max: 8)
- Average Confidence: 4 (Min: 4, Max: 4)

Reviews highlighted the following strenghts:
- clear written presentation (R1+R3)
- related work well covered (R1+R3) and built upon (R2)
- supplementary material detailed (R1) and available (R3)
- insight on pros/cons of evalutation techniques provided (R1)
- chosen baseline technique (Digits) is realistic (R2)

Reviews pointed the following weaknesses:
- challenging visual presentation and positioning of figures and/versus related text (R1+R2+R3)
- references (R2) or details (R3) missing for some claims: particularly on large displays
- weak research question framing (R1)
- lack of design choice rationale (R1)

I recommend for acceptance of this paper.

---

### Decision · Program_Chairs · 2020-04-25

Accept